# Microbial Biosynthesis of Chrysazin Derivatives in Recombinant *Escherichia coli* and Their Biological Activities

**DOI:** 10.3390/molecules27175554

**Published:** 2022-08-29

**Authors:** Purna Bahadur Poudel, Dipesh Dhakal, Rubin Thapa Magar, Jae Kyung Sohng

**Affiliations:** 1Institute of Biomolecule Reconstruction (iBR), Department of Life Science and Biochemical Engineering, Sun Moon University, 70 Sun Moon-ro 221, Tangjeong-myeon, Asan-si 31460, Chungnam, Korea; 2Department of Biotechnology and Pharmaceutical Engineering, Sun Moon University, 70 Sun Moon-ro 221, Tangjeong-myeon, Asan-si 31460, Chungnam, Korea

**Keywords:** rhamnosyltransferase, *O*-methyltransferase, chrysazin, biological properties

## Abstract

Anthraquinone and its derivatives show remarkable biological properties such as anticancer, antibacterial, antifungal, and antiviral activities. Hence, anthraquinones derivatives have been of prime interest in drug development. This study developed a recombinant *Escherichia coli *strain to modify chrysazin to chrysazin-8-*O*-*α*-l-rhamnoside (CR) and chrysazin-8-*O*-α-l-2′-*O*-methylrhamnoside (CRM) using rhamnosyl transferase and sugar-*O*-methyltransferase. Biosynthesized CR and CRM were structurally characterized using HPLC, high-resolution mass spectrometry, and various nuclear magnetic resonance analyses. Antimicrobial effects of chrysazin, CR, and CRM against 18 superbugs, including 14 Gram-positive and 4 Gram-negative pathogens, were investigated. CR and CRM exhibited antimicrobial activities against nine pathogens, including methicillin-resistant *Staphylococcus aureus* (MRSA) and methicillin-sensitive *Staphylococcus aureus* (MSSA) in a disk diffusion assay at a concentration of 40 µg per disk. There were MIC and MBC values of 7.81–31.25 µg/mL for CR and CRM against methicillin-sensitive *S. aureus* CCARM 0205 (MSSA) for which the parent chrysazin is more than >1000 µg/mL. Furthermore, the anti-proliferative properties of chrysazin, CR, and CRM were assayed using AGS, Huh7, HL60, and HaCaT cell lines. CR and CRM showed higher antibacterial and anticancer properties than chrysazin.

## 1. Introduction

Quinones are naturally occurring organic compounds found in higher plants, fungi, bacteria, and animals. They have a lot of structural varieties. Since they are found in different colors in nature, they are considered pigments [1]. Anthraquinones are the largest group of quinones, with various biological properties such as antioxidant [2], antifungal [3], antiviral [4], anti-diabetic [5], anti-inflammatory [6], and laxative [7] effects. They are also used as natural dyes in industries [8]. Several anthraquinones are widely used in the treatment of cancer [9,10,11]. They show cytotoxic activities through interaction with DNA, preferentially at cytosine/guanine-rich sites [12].

Synthesis of anthraquinone derivatives is of great interest recently. There are various methods for the synthesis of anthraquinones derivatives, including intramolecular condensation of aryl and o-aroylbenzoic acid using fuming sulfuric acid, benzoyl chloride, concentrated sulfuric acid, benzoyl chloride, zinc chloride, and POCl_3_/P_2_O_3_Cl_4_ [13]. The chemical synthesis of anthraquinones derivatives is an expensive and difficult process [14].

Modification of anthraquinones can be performed by glycosylation, methylation, sulfation, prenylation, and so on. Glycosylation is an important process for increasing the solubility of hydrophobic compounds, improving stability, reducing toxicity, and modifying biological activities [15,16]. Methylation can alter the solubility, deactivate the reactive hydroxyl group, increase the metabolic stability, increase membrane transport, and increase pharmaceutical properties [17,18,19].

Methyltransferases are important enzymes in the modification of different substrate. However, methylation of sugar is very rare [20,21]. Generally, transfer of the sugar group is catalyzed by glycosyltransferases (GTs) with activated NDP-sugars as sugar donors. Methylation reactions are catalyzed by *O*-methyltransferase (OMT) that catalyzes the transfer of the methyl group of *S*-adenosyl-l-methionine (SAM) to hydroxyl groups [15,22,23].

Chrysazin (Dantron; 1,8 dihydroxyanthraquinone) has been used as a medicine since ancient times. It can be found naturally. It is isolated from the root and rhizome of *Rheum palmatum L.* (Polygonaceae) [24]. It has a wide range of activities, such as antitumor activity, confirmed by different experiments [24,25].

This study is based on the utilization of indigenous *E. coli* sugar (thymidine diphosphate (TDP)-rhamnose) as a sugar donor, which will be used as a glycosyltransferase to conjugate to chrysazin and SAM as an *O*-methyltransferase to conjugate chrysazin rhamnoside. For the production of chrysazin derivatives, anthraquinone rhamnosyltransferase (7665) [26] from *Saccharothrix espanaensis* and *O*-methyltransferase (ThnM1) [27] from *Nocardia* sp. CS682 were cloned and heterologously expressed in *E. coli*. The recombinant strain *E. coli* was utilized for the production of chrysazin-8-*O*-*α*-l-rhamnoside (CR) and chrysazin-8-*O*-α-l-2′-*O*-methylrhamnoside (CRM) from the substrate chrysazin (Figure 1). Anticancer and antibacterial activities of CR and CRM were assessed and results were compared to those of chrysazin.

## 2. Materials and Methods

### 2.1. General Procedures

Chrysazin/Dantron was purchased from Tokyo Chemical Industry (Tokyo, Japan). HPLC-grade acetonitrile and water were purchased from Mallinckrodt Baker (Phillipsburg, NJ, USA). All other chemicals used were of high analytical grade and commercially available. Isopropyl-*β*-d-1-thiogalactoside (IPTG) was purchased from GeneChem Inc. (Daejeon, Korea). SAM was purchased from Sigma-Aldrich (St. Louis, MO, USA). *Escherichia coli* BL21 (DE3) (Stratagene, La Jolla, CA, USA) was used as an expression and biotransformation host. Luria–Bertani (LB) broth medium and agar plates with appropriate antibiotics (ampicillin, kanamycin, chloramphenicol, and streptomycin, each at 50 μg/mL) were used for culture preparation, colony selection, and biotransformation. Pathogenic strains such as *Staphylococcus. aureus* CCARM 3640 (MRSA), *S. aureus* CCARM 3089 (MRSA), *S. aureus* CCARM 33591(MRSA), *S. aureus* CCARM 0205 (MSSA), *S. aureus* CCARM 0204 (MSSA), *S. aureus* CCARM 0027 (MSSA), *S. aureus* CCARM 3090 (MRSA), *S. aureus* CCARM 3634 (MRSA), *S. aureus* CCARM 3635 (MRSA), *Bacillus subtilis* ATCC 6633, *Enterococcus faecalis* 19433, *Enterococcus faecalis* 19434, *Kocuria rhizophilla* NBRC 12708, *Micrococcus luteus*, *Escherichia coli* ATCC 25922, *Proteus hauseri* NBRC 3851, *Klebsiella pneumonia* ATCC10031, and *Salmonella enterica* ATCC 14,028 were obtained from Professor Seung-Young Kim (Sun Moon University, Korea) [28].

### 2.2. Generation of Recombinant Strains

For the production of rhamnosylated and methoxy-rhamnosylated derivatives of chrysazin using engineered *E. coli*, anthraquinone rhamnosyltransferase (7665) [29] from *S. espanaensis* and rhamnose methyltransferase (ThnM1) [27] from *Nocardia* sp. CS682 were taken to prepare *E*. *coli* S2 [30]. *E. coli* S2 was generated by transforming pET32a ± 7665(Am) CRISPRi-S1(Cm^r^) and pCDFDuet-metK-thnM1(Sm) into an *E. coli* strain harboring the rhamnose cassette piBR181-tgs.dh.ep.kr.pgm2.glf.glk (Km) [31]. Recombinant plasmids were confirmed by restriction endonuclease activity as well as by growing in a combined four-antibiotic LB agar plate and LB broth medium. 

### 2.3. Culture Preparation and Whole-Cell Biotransformation

A seed culture of *E. coli* S2 was cultured in a 5 mL LB medium supplemented with ampicillin, kanamycin, chloramphenicol, and streptomycin (each at 50 μg/mL) and incubated at 37 °C for 3 h. From the 5 mL seed culture, 500 µL was transferred into a flask containing 50 mL LB broth with respective antibiotics and cultured at 37 °C for around 4 h until the optical density of cells at 600 nm (OD_600_) reached 0.6–0.8. This culture had added to it 0.5 mm isopropyl β-d-1-thiogalactopyranoside (IPTG) to induce protein expression, followed by incubation at 20 °C for 20 h. To determine optimal substrate concentration, glucose concentration, and time, different concentrations (1, 2, 4, 6, 8, 10 mm) of chrysazin and different concentrations (2%, 5%, 10%, 12%, and 15%) of sterile glucose solution were fed into the cell culture after induction. After 20 h of adding IPTG, chrysazin dissolved in dimethyl sulfoxide (DMSO) at a final concentration of 400 µM was added along with 10% glucose. Samples (3 mL of culture broths) were withdrawn at 6, 12, 24, 36, 48, and 60 h for product analysis. After 48 h, compounds were extracted using a double volume of ethyl acetate (2:1, *v*/*v*) and a Soxhlet extractor. The Soxhlet extractor was kept still to separate the mixture for 4–6 h at room temperature after shaking. Ethyl acetate was removed under reduced pressure and dissolved in methanol. This sample was further analyzed by HPLC and mass spectrometry. To collect a sample to characterize structurally, the biotransformation experiment was performed using a fermenter (3 L of culture). The pure fraction of the compound was collected via preparatory-high-pressure liquid chromatography (prep-HPLC).

### 2.4. Analytical Procedures

From the extracted compound, a 20 µL volume was injected and directly analyzed by reverse-phase high-performance liquid-chromatography photo-diode array (HPLC-PDA) using a Thermo Scientific Dionex Ultimate 3000 ultrahigh-performance Liquid chromatography (UHPLC) system with a reverse-phase C_18_ column (Mightysil RP-18 GP (4.6 mm × 250 mm, 5 μm particle size) (Kanto Chemical, Tokyo, Japan)). The binary mobile phase was composed of solvent A (HPLC-grade water + 0.1% trifluoroacetic acid) and solvent B (100% acetonitrile, ACN). The total flow rate was maintained at 1 mL/min for the 30 min program. The ACN concentration began with 10%. A linear gradient from 10 to 50% for 10 min, 50–90% for 23 min, and 90–10% for 30 min was then used. The HR-QTOF ESI/MS was performed in positive ion mode using an Acquity mass spectrometer (Waters, Milford, MA, USA), which was coupled with a Synapt G2-S system (Waters). Purification of compounds was performed using a prep-HLPC instrument equipped with a YMC-Pack ODS-AQ C_18_ column, (150 × 20 mm I.D., mean particle size: 10 μm) (YMC America, Inc., Devens, MA, USA) and a connected UV detector (420 nm). Here, a 40 min binary program with implementation of 20% (0–5 min), 50% (5–10 min), 70% (10–20 min), 90% (20–25 min), 20% (25–30 min), and 10% (30–35 min) ACN at a flow rate of 10 mL/min was used. Purified products were pooled, dried, and lyophilized to remove water or moisture. Furthermore, the fully dried pure compound was dissolved in DMSO-*d6* and subjected to a 700 MHz NMR spectrometer equipped with TCI CryoProbe (5 mm). 

One-dimensional NMRs (^1^H NMR and ^13^C NMR) and two-dimensional NMRs (heteronuclear multiple quantum coherence (HMQC), rotating frame Overhauser enhancement spectroscopy (ROESY), and heteronuclear multiple bonded connectivity (HMBC)) were used as needed to elucidate the structure of the compound.

### 2.5. Anticancer Activities of Chrysazin, CR, and CRM

Three cancer cell lines (i.e., human liver cancer cell line (Huh7), human gastric cancer cell line (AGS), and human leukemia cell line (HL60) and normal cell line (human keratinocyte cell line (HaCaT) were purchased from Korean Cell Line Bank (Seoul, Korea). Huh7 cells were cultured in Dulbecco’s modified Eagle’s medium (DMEM) (Corning Cellgro Manassas, VA, USA) and AGS cells were cultured in Roswell Park Memorial Institute 1640 medium (RPMI1640) (Corning Cellgro, Manassas, VA, USA) supplemented with 10% fetal bovine serum (FBS) (Grand Island, NY, USA) and 1% penicillin-streptomycin-amphotericin B (Walkersville, MD, USA). Human leukemia HL60 cells were cultured in RPMI1640 supplemented with 10% FBS, 1% penicillin-streptomycin-amphotericin B, and L-glutamine (2 mm) (Grand Island, NY, USA). HaCaT cell lines were grown in DMEM supplemented with 10% FBS, 100 μg/mL streptomycin, and 100 μg/mL benzylpenicillin. All cells were maintained at 37 °C in a humidified 5% CO_2_ incubator. For cell growth assay, cells were seeded at 3 × 10^2^ cells/well into white 96-well culture plates (SPL Life Sciences, Pochon, Korea), incubated at 37 °C in a humidified 5% CO_2_ overnight, and then treated with each compound after serial dilution (200 μM, 100 μM, 50 μM, 25 μM, 12.5 μM, 6.25 μM, 3.16 μM, 1.56 μM, 0.78 μM) for 72 h. After that, 20 μL substrate solution (Promega) was added to each well. The plate was shaken for 5 min and kept in the dark for 10 min. Luminescence was measured using a multimode plate reader (BioTek, Inc., Winooski, VT, USA). IC_50_ values were analyzed using GraphPad Prism 5 (GraphPad Software, La Jolla, CA, USA).

### 2.6. Antimicrobial Activities of Chrysazin, CR, and CRM

#### 2.6.1. Disk Diffusion Assay

Fourteen Gram-positive bacteria (*Staphylococcus. Aureus* CCARM 3640 (MRSA), *S. aureus* CCARM 3089 (MRSA), *S. aureus* CCARM 33591(MRSA), *S. aureus* CCARM 0205 (MSSA), *S. aureus* CCARM 0204 (MSSA), *S. aureus* CCARM 0027 (MSSA), *S. aureus* CCARM 3090 (MRSA), *S. aureus* CCARM 3634 (MRSA), *S. aureus* CCARM 3635 (MRSA), *Bacillus subtilis* ATCC 6633, *Enterococcus faecalis* 19433, *Enterococcus faecalis* 19434, *Kocuria rhizophilla* NBRC 12708, and Micrococcus luteus) and four Gram-negative bacteria (*Escherichia coli* ATCC 25922, *Proteus hauseri* NBRC 3851, *Klebsiella pneumonia* ATCC10031, and *Salmonella enterica* ATCC 14028) were used to test antibacterial activities of chrysazin and its derivatives. The paper disk diffusion assay on the Mueller–Hinton agar (MHA) plate was carried out according to Clinical Laboratory Standard Institute (CLSI) guidelines and the Kirby–Bauer method [32,33]. Inocula containing 10^8^ colony forming units (CFU)/mL were spread onto MHA plates. Then, 40 µg/disk compounds were placed on the surface of inoculated agar plates using sterile paper disks of 6 mm (Advantec, Toyo Roshi Kaisha, Ltd., Japan). Samples were then incubated at 37 °C for 18–20 h. The zone of inhibition diameter was measured in millimeters for each pathogen. Dimethyl sulfoxide (DMSO) was used as a control for the zone of inhibition as all compounds were dissolved in DMSO.

#### 2.6.2. Measurements of MIC and MBC of Chrysazin Derivatives

The following nine strains were used in the minimum inhibitory concentration (MIC) and minimum bactericidal concentration (MBC) tests: *Staphylococcus. aureus* CCARM 3640 (MRSA), *S. aureus* CCARM 3089 (MRSA), *S. aureus* CCARM 33591(MRSA), *S. aureus* CCARM 0205 (MSSA), *S. aureus* CCARM 0204 (MSSA), *S. aureus* CCARM 0027 (MSSA), *S. aureus* CCARM 3090 (MRSA), *S. aureus* CCARM 3634 (MRSA), and *S. aureus* CCARM 3635 (MRSA). They were grown in Mueller–Hinton Broth (MHB) (Difco, Baltimore, MD, USA). The broth dilution method was used to determine MIC [34]. The MHB and sample were dispensed in a 96-well plate and serially diluted. The strain was inoculated into each well and cultured for 16–20 h at 37 °C. Each strain’s suspension was adjusted to 0.5 McFarland standard (1 × 10^8^ CFU/mL) and then diluted to 2.5 × 10^6^ CFU/mL in MHB. After knowing the MIC, the MBC test was performed on a fresh MHB medium by inoculating cultured samples containing MIC compounds and experimental strains.

## 3. Results and Discussion

### 3.1. Biosynthesis of CR and CRM

The recombinant strain of *E. coli* strain S2 was generated by engineering *E. coli* BL21 (DE3), which contained a sugar transfer cassette and a sugar methylation cassette. It was cultured and prepared for biotransformation as mentioned in Section 2.2. For optimal production, different concentrations (1, 2, 4, 6, 8, and 10 mm) of chrysazin and different concentrations (2%, 5%, 10%, 12%, and 15%) of glucose were tested with different time intervals (6, 12, 24, 36, 48, and 60 h). It was observed that 400 µM, 10% glucose, and 48 h were suitable conditions (Appendix A). The biotransformation system was induced with 0.5 mm of IPTG. After 20 h, 400 µM of chrysazin and 10% glucose were supplied into cell cultures. The extract from engineered *E. coli* strain S2 was analyzed by HPLC. The HPLC chromatogram of chrysazin was obtained with its standard retention time (tR) of 21.3 min. Two new peaks of CR and CRM were obtained at tR of 14.9 min and 16.3 min, respectively, with UV absorbance at 420 nm (Figure 2). The reaction mixture was further analyzed by high-resolution quadrupole time-of-flight electrospray ionization mass spectrometry (HR-QTOF ESI/MS). The product mass fragment of CR [M + H]^+^ *m*/*z* = 387.1047 was matched to the calculated mass of CR [M + H]^+^ *m*/*z* = 387.1074. Similarly, the product mass fragment of CRM [M + Na]^+^ *m*/*z* = 423.1057 was matched to the calculated mass [M + Na]^+^ *m*/*z* = 423.1056 in the positive ion mode (Figure 3), which resembled the rhamnosylated and rhamnose-methylated derivatives of chrysazin.

### 3.2. Purification and Structural Elucidation of the Metabolite

Biotransformation was carried out through fermentation to collect CR and CRM for structure identification and further biological activity tests. The biotransformation reaction mixture was extracted with a double volume of ethyl acetate. The crude extract was subjected to preparatory-high-pressure liquid chromatography (prep-HPLC) for purification. After several rounds of prep-HPLC, purified compounds were obtained. The purified product was dried by lyophilization, dissolved in 400 µL of deuterated dimethyl sulfoxide, and analyzed by nuclear magnetic resonance (NMR) spectroscopy (700 MHz) including 1D NMR (^1^H-NMR and ^13^C-NMR) and 2D NMR (HMBC, HSQC, COSY, and ROESY), as shown in Appendix A, and Table 1 for structural elucidation.

The ^1^H-NMR of chrysin, CR, and CRM showed multiple peaks between 1.0 ppm and 13.0 ppm. In the case of CR, the rhamnose group was attached to the 8-OH group of chrysazin. The anomeric proton (1′ -H) was consistent with δ 5.67 (d, J = 1.1 Hz, 1H), in which the anomeric proton coupling constant (J = 1.1 Hz) confirmed that the conjugation of rhamnose moiety had an α-configuration. In addition, with ^13^C-NMR of CR, the anomeric carbon peak appeared at δ 99.08 ppm, with other peaks appearing between 70 and 80 ppm along with a CH_3_ peak at 18.35 ppm. In the case of CRM, there was a methylation in the 2′ -OH group of rhamnose in CR, where the OCH_3_ spectrum was visible in both ^1^H and ^13^C NMR at 3.5 ppm and 59.44 ppm, respectively. Furthermore, to confirm the sugar and sugar-*O*-methylation conjugation, two-dimensional (2D)-NMR analyses such as ^1^H-^13^C HSQC, ^1^H-^13^C HMBC, ^1^H-^1^H COSY, and ^1^H-^1^H ROESY experiments were performed. Similarly, in CR, HSQC showed a cross peak illustrating a correlation between the anomeric C-1′ proton (*δ* 5.67 ppm) and the anomeric carbon (*δ* 99.00 ppm). Moreover, the C-8 signal appearing at *δ* 157.18 ppm showed a direct correlation with the observed anomeric proton at *δ* 5.67 ppm in HMBC (Appendix A). In the case of CRM, HSQC showed a cross peak illustrating a correlation between the C-2′ protons (*δ* 3.71 ppm) and the carbon (*δ* 80.61 ppm), and HMBC showed a cross peak depicting the correlations between C-2′ (*δ* 80.62 ppm) and the protons of the methoxy group (*δ* 3.50 ppm) (Appendix A). Results shown above reveal that the glycosylated derivative of chrysazin (i.e., the chrysazin-8-*O*-*α*-l-rhamnoside) and methylated derivative of CR (i.e., chrysazin-8-*O*-α-l-2′-*O*-methylrhamnoside) produced by *E. coli* S2 whole-cell biotransformation were new compounds.

### 3.3. Anticancer Activities

The three compounds prepared were further analyzed for their in vitro cytotoxicities using 3-(4,5-dimethylthiazol-2-yl)-2,5- diphenyltetrazolium bromide (MTT) colorimetric assay against three different cancer cell lines (Figure 4) and one normal cell line (Appendix A). Two derivatives of chrysazin (i.e., CR and CRM) showed higher cytotoxicities than chrysazin. The 50% inhibitory concentration (IC_50_) values of chrysazin for AGS, Huh7, and HL60 cells were 17.08, 30.53, and 22.24 (μM), respectively. Chrysazin-8-*O*-*α*-l-rhamnoside inhibited AGS, Huh7, and HL60 cells with 50% inhibitory concentration (IC_50_) values of 28.58, 21.28, and 14.68 (μM), respectively. Here, CR showed better anticancer activities in Huh7 and HL60 than chrysazin. In the case of AGS cells, it showed slightly lower anticancer activity. In the case of chrysazin-8-*O*-α-l-2′-*O*-methylrhamnoside, it showed higher anticancer activities than chrysazin and CR. The 50% inhibitory concentration (IC_50_) values of CRM for AGS, Huh7, and HL60 cells were 7.513, 4.467, and 4.540 (μM), respectively. In the case of the HaCaT normal cell line, the IC_50_ values were >200 μM for chrysazin, CR, and CRM (Appendix A and Appendix A). These compounds had a lower inhibitory effect on normal cells. These results suggest that CR and CRM can remarkably reduce cell viabilities of AGS, Huh7, and Hl60 cells in a dose-dependent manner. This is the first report of the activity of these two compounds against AGS, Huh7, and HL60 cells.

### 3.4. Antimicrobial Bacterial Activities

#### 3.4.1. Disk Diffusion Assay

Paper-disk diffusion assay was performed to determine the antimicrobial activity. All three compounds (chrysazin, CR, and CRM) were prepared at a concentration of 10 mg/mL. All compounds were added to each disk at a final concentration of 40 µg/disk (4 µL). Each disk was placed over Mueller–Hinton agar (MHA) plates spread with bacterial strains. The diameter of the zone of inhibition was measured after 18–20 h. Results of disk diffusion assays revealed that chrysazin did not show any antibacterial activity against 18 different human pathogens tested. However, CR and CRM exhibited antibacterial activities against Gram-positive bacteria *S. aureus* subsp. (Appendix A and Appendix A). These results reveal that rhamnosylation and rhamnose methylation of chrysazin might be profitable for heightening its antibacterial activity against different Gram-positive bacteria.

#### 3.4.2. Measurement of MIC and MBC Values

Minimum inhibitory concentration (MIC) and minimum bactericidal concentration (MBC) for chrysazin, chrysazin-8-*O*-*α*-l-rhamnoside, and chrysazin-8-*O*-α-l-2′-*O*-methylrhamnoside against nine different pathogenic bacteria were determined. MIC values of compound chrysazin, CR, CRM, and erythromycin as a positive control, against nine strains of Gram-positive bacteria *S. aureus* subsp were determined by the broth dilution method. The assays were performed in 96-well plates in duplicate with Mueller–Hinton broth. As summarized in Table 2, CR and CRM exhibited antibacterial activity against *S*. *aureus* CCARM 0205 (MSSA), *S. aureus* CCARM 0204 (MRSA), *S. aureus* CCARM 3640 (MRSA), *S. aureus* CCARM 3090 (MRSA), *S. aureus* CCARM 3634 (MRSA), *S. aureus* CCARM 0027 (MSSA), *S. aureus* CCARM 3089 (MRSA), *S. aureus* CCARM 3635 (MRSA), and *S. aureus* CCARM 33591(MRSA), with MIC values of 7.81–1000 µg/mL. After knowing the MIC value, we further analyzed MBC, the lowest concentration of a compound for killing inoculated bacteria. An MBC test was performed by inoculating cultured samples containing MIC compounds and experimental strains on a fresh MHB medium (Table 3). MBC values were similar to or higher than MIC values in all tested strains. Chrysazin did not show antibacterial activity against different pathogenic bacteria but derivatives of it, i.e., CR and CRM, improve its antibacterial activity. The metabolite showed significantly improved antibacterial activity against a wide range of Gram-positive MRSA and MSSA pathogens. Therefore, CR and CRM have great potential as an antibiotic against superbugs.

## 4. Conclusions

In this study, we successfully engineered an *E. coli* strain for the sustainable production of different derivatives of chrysazin. Chrysazin-8-*O*-*α*-l-rhamnoside and chrysazin-8-*O*-α-l-2′-*O*-methylrhamnoside were novel compounds. We also evaluated their activities against three different cancer cell lines (AGS, Huh7, and HL60). CR and CRM exhibited higher cytotoxicities than their parental compound, chrysazin. More significantly, the evaluation of antibacterial activity revealed promising bioactivities of CR and CRM. This study establishes an engineered microbial platform that can be used to produce novel bioactive compounds. This microbial platform can be further fine-tuned for the production of novel derivatives of other anthraquinones or different compounds. Furthermore, the optimization of bioprocessing parameters and rational engineering of the host using various synthetic biological tools and metabolic engineering can be employed to enhance the production titer.

## Figures and Tables

**Figure 1 molecules-27-05554-f001:**
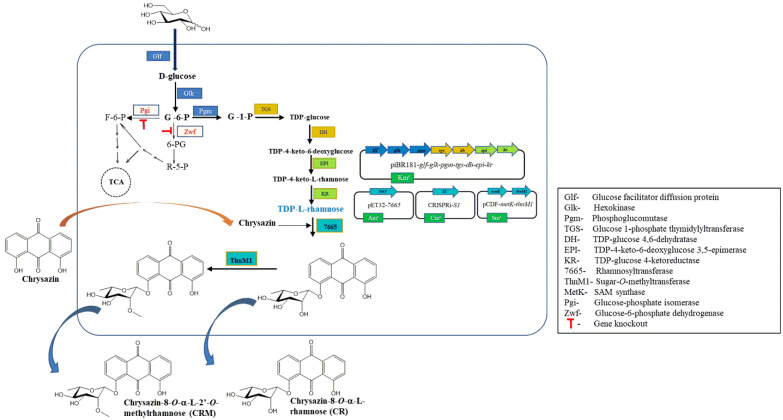
A scheme showing the pathway of utilizing recombinant *Escherichia coli* BL21(DE3) for the biosynthesis of chrysazin-8-*O*-*α*-l-rhamnoside (CR) and chrysazin-8-*O*-α-l-2′-*O*-methylrhamnoside (CRM) from chrysazin. Chromosomal *pgi* (glucose-phosphate isomerase) and *zwf* (glucose-6-phosphate dehydrogenase) genes were knocked-out. Chromosomal *glk* (hexokinase), *pgm* (phosphoglucomutase), *tgs* (glucose 1-phosphate thymidylyltransferase), *dh* (TDP-glucose 4,6-dehydratase), *epi* (TDP-4-keto-6-deoxyglucose 3,5-epimerase), and *kr* (TDP-glucose 4-ketoreductase) genes were overexpressed by cloning into pIBR181. *7665* (rhamnosyl transferase) was overexpressed by cloning into pET32a (+), and *metK* (SAM synthase) and *thnM1* (sugar-*O*-methyltransferase) were overexpressed by cloning into pCDF-Duet.

**Figure 2 molecules-27-05554-f002:**
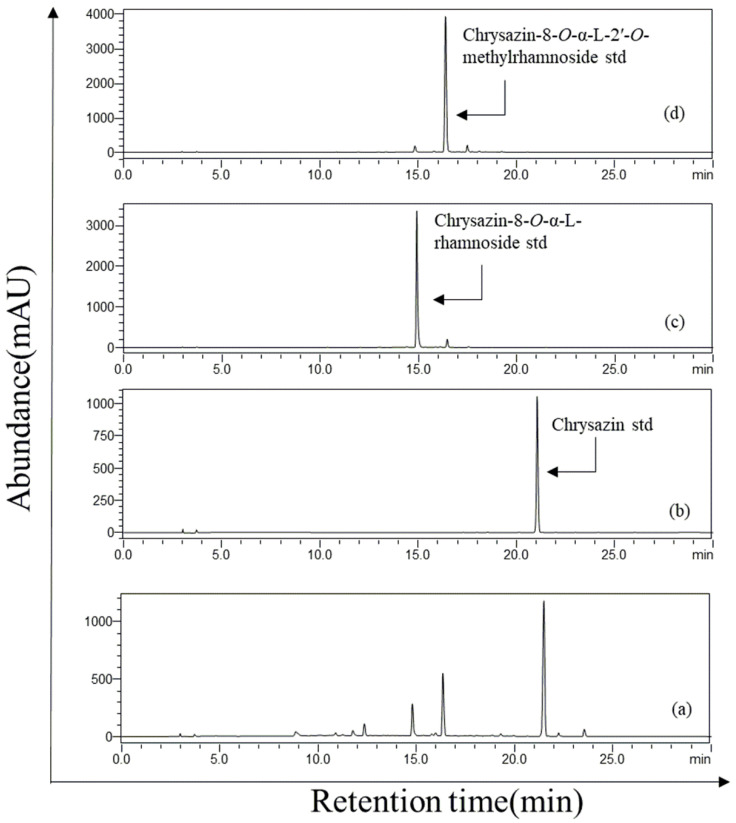
Whole-cell biotransformation of chrysazin to chrysazin-8-*O*-*α*-l-rhamnoside and chrysazin-8-*O*-α-l-2′-*O*-methylrhamnoside using engineered *E. coli* S2 overexpressing anthraquinone glycosyltransferase, sugar-MT (ThnM1), TDP-rhamnose sugar biosynthetic pathway overexpressing plasmid, and SAM synthase overexpressing plasmid. HPLC-PDA chromatogram analyses of (**a**) biotransformation reaction sample compared to (**b**) chrysazin standard, (**c**) chrysazin-8-*O*-*α*-l-rhamnoside standard, (**d**) chrysazin-8-*O*-α-l-2′-*O*-methylrhamnoside.

**Figure 3 molecules-27-05554-f003:**
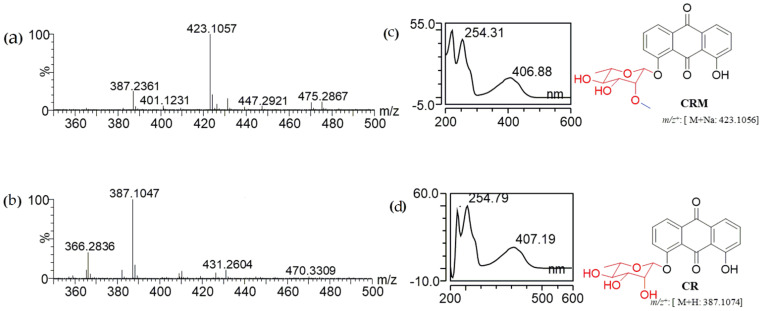
HR-QTOF ESI/MS chromatogram of (**a**) chrysazin-8-*O*-α-l-2′-*O*-methylrhamnoside; (**b**) chrysazin-8-*O*-*α*-l-rhamnoside; (**c**) UV/VIS of CRM, (**d**) UV/VIS of CR.

**Figure 4 molecules-27-05554-f004:**
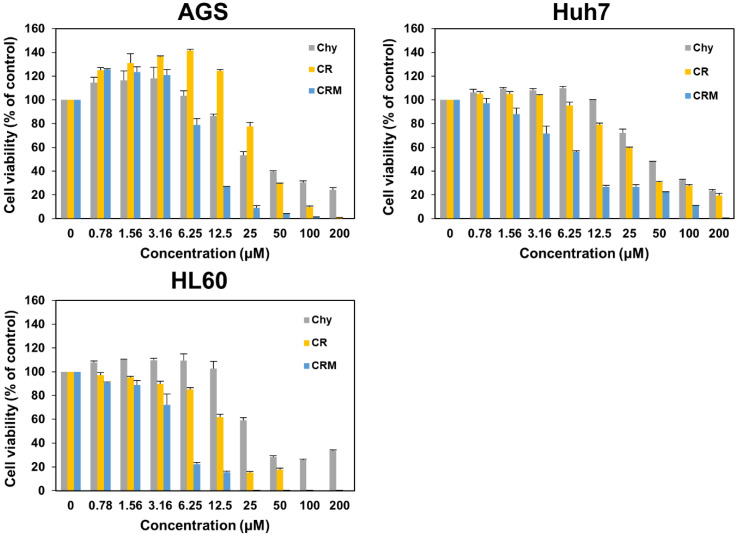
Cell cytotoxicity assay results of chrysazin, chrysazin-8-*O*-*α*-l-rhamnoside, and chrysazin-8-*O*-α-l-2′-*O*-methylrhamnoside. Cells were treated with various concentrations (0.0–200 μM) of each compound.

**Table 1 molecules-27-05554-t001:** Comparison of ^1^H- and ^13^C-NMR chemical shifts of chrysazin, chrysazin-8-O-α-l-rhamnoside (CR), and chrysazin-8-O-α-l-2′-O-methylrhamnoside (CRM) measured in DMSO-d6 solvent.

^1^H-NMR (700 MHz, DMSO-*d6*)				^13^C-NMR (176 MHz, DMSO-*d6*)		
Position	Chrysazin	CR	CRM	Chrysazin	CR	CRM
1				161.35	161.94	161.94
2	7.37 (dd, *J* = 8.4 Hz, 1H)	7.64 (dd, *J* = 7.5 Hz, 1H)	7.32 (d, *J* = 8.5 Hz, 1H)	124.44	118.81	124.67
3	7.8 (m, 1H)	7.74 (dd, *J* = 7.9 Hz, 1H)	7.71 (m, 1H)	137.48	136.80	123.72
4	7.70 (dd, *J* = 7.5 Hz, 1H)	7.35 (dd, *J* = 8.3 Hz,1H)	7.60 (d, *J* = 7.3 Hz, 1H)	119.33	124.69	118.80
4a				133.29	132.87	132.79
5	7.70 (dd, *J* = 7.5 Hz, 1H)	7.85 (d, *J* = 2.3 Hz, 1H)	7.84 (m*,* 1H)	119.33	136.42	136.34
6	7.8 (m, 1H)	7.85 (s, 1H)	7.70 (d, *J* = 13.3 Hz, 1H))	137.48	121.02	136.78
7	7.37 (dd, *J* = 8.4 Hz, 1H)	7.69 (ddd, *J* = 9.6 Hz, 1H)	7.84 (m, 1H)	124.44	123.31	121.21
8	7.99 (dd, *J* = 5.79, 3.31 Hz, 1H)			161.35	157.27	157.19
8a				115.93	121.39	121.49
9				192.02	188.63	188.57
9a				115.93	117.21	117.15
10				181.37	182.35	188.90
10a				133.29	135.47	135.39
11	11.90 (s, 1H)	12.98 (s, 1H)	12.94 (s, 1H)			
12	11.90 (s, 1H)					
1’		5.67 (d, *J* = 1.1 Hz, 1H)	5.84 (s, 1H)		99.08	96.25
2’		4.01 (m, 1H),	3.71 (m, 1H),		70.54	80.65
3’		4.01 (m, 1H)	4.09 (m, 1H)		70.59	70.48
4’		3.35 (ddd, *J* = 9.3 Hz, 1H)	3.29 (m, 1H)		72.15	72.48
5’		3.51 (m, 1H)	3.51 (s, 1H)		70.66	70.46
6’-CH_3_		1.10 (d, *J* = 6.2 Hz, 3H)	1.10 (d, *J* = 6.3 Hz, 3H)		18.35	18.33
12-O-CH_3_			3.5 (s, 3H)			59.44

Coupling constant is represented as *J*, whereas multiplicities are indicated by s (singlet), d (doublet), and m (multiplet), and the chemical shift values are in ppm.

**Table 2 molecules-27-05554-t002:** MIC values of compound chrysazin, CR, CRM, and erythromycin (Erm) against 9 strains.

MIC (µg/mL)				
	Chrysazin	CR	CRM	Erm
*S. aureus* CCARM 0205 (MSSA)	>1000	15.62	7.81	3.91
*S. aureus* CCARM 0204 (MRSA)	>1000	62.5	15.62	3.91
*S. aureus* CCARM 3640 (MRSA)	>1000	250	62.5	>1000
*S. aureus* CCARM 3090 (MRSA)	>1000	250	125	500
*S. aureus* CCARM 3634 (MRSA)	>1000	250	125	>1000
*S. aureus* CCARM 0027 (MSSA)	>1000	500	250	3.91
*S. aureus* CCARM 3089 (MRSA)	>1000	1000	500	>1000
*S. aureus* CCARM 3635 (MRSA)	>1000	1000	500	500
*S. aureus* CCARM 33591(MRSA)	>1000	>1000	500	>1000

Erm (erythromycin): positive control.

**Table 3 molecules-27-05554-t003:** MBC values of compound chrysazin, CR, CRM, and erythromycin (Erm) against 9 strains.

MBC (µg/mL)				
	Chrysazin	CR	CRM	Erm
*S. aureus* CCARM 0205 (MSSA)	>1000	31.25	15.62	7.81
*S. aureus* CCARM 0204 (MRSA)	>1000	125	15.62	7.81
*S. aureus* CCARM 3640 (MRSA)	>1000	250	125	>1000
*S. aureus* CCARM 3090 (MRSA)	>1000	500	125	500
*S. aureus* CCARM 3634 (MRSA)	>1000	500	250	>1000
*S. aureus* CCARM 0027 (MSSA)	>1000	500	250	7.81
*S. aureus* CCARM 3089 (MRSA)	>1000	>1000	1000	>1000
*S. aureus* CCARM 3635 (MRSA)	>1000	1000	1000	1000
*S. aureus* CCARM 33591(MRSA)	>1000	>1000	1000	>1000

Erm (erythromycin): positive control.

## Data Availability

Not applicable.

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
