# Peer review of "Microbial Biosynthesis of Chrysazin Derivatives in Recombinant Escherichia coli and Their Biological Activities"

_molecules, 2022, doi:10.3390/molecules27175554_

Round 1

Reviewer 1 Report

The article is fluent and logical. The CR and CRM showed high antibacterial and anticancer properties. But whether there should have a toxicity test on human normal cells in the cytotoxic activity test. And there is no Supporting Information in the submitted file, so there is no NMR correlation of the compounds.

Author Response

The article is fluent and logical. The CR and CRM showed high antibacterial and anticancer properties. But whether there should have a toxicity test on human normal cells in the cytotoxic activity test. And there is no Supporting Information in the submitted file, so there is no NMR correlation of the compounds

Answer:  Thank you very much for your positive comments. According to the reviewer’s suggestion, we tested the cytotoxicity assay on non-tumor cell lines, and we found these compounds had a lower inhibitory effect on normal cells (Table S1 and Figure S6). An appropriate correction has been duly made in the main manuscript and COSY, ROSEY, and HMBC in the supplementary data.

Cell Lines

Chrysazin (Chy)

CR

CRM

Cancer cell lines

AGS

17.08

28.58

7.513

Huh7

30.53

21.28

4.467

HL60

22.24

14.68

4.540

Normal cell lines

HaCaT

>200

>200

>200

Table S1. Anticancer (IC50 (mM)) potential of chrysazin analogues against different cancer cell lines.

Reviewer 2 Report

The study “Microbial biosynthesis of chrysazin derivatives in recombinant Escherichia coli and their biological activities” is interesting but I have several doubts about some experiments.

1)Please, uniform anticancer in the text. At some point is present anticancer in other anti-cancer.

2)Please, rephrase “In this study, a recombinant Escherichia coli strain was developed to modify chrysazin” in “This study developed a recombinant Escherichia coli strain to modify chrysazin”.

3) please, uniform anti-microbial in the text. In some points is present antimicrobial in other anti-microbial.

4)Please, change “Disc” to “disk”.

5)Generally, the disk diffusion assay is performed following EUCAST or CLSI guidelines. Is not clear which kind of protocol the authors used. Please, add a reference.

6)In the disk diffusion assay the authors used inocula containing 107 colony forming units (CFU)/mL. The guidelines recommend to use 0.5 McFarland which corresponds to 1.5 x10CFU/mL and measuring after 18-20h. Why in the method of micro dilutions and agar the result was evaluated at different times, 12 and 20h?

7) Figure s5 is not visible. I can not appreciate the measures of the diameter area.

8)erythromycin was used as a positive control in all antibacterial tests. All MRSA strains were sensitive to this antibiotic? Are antibiograms available?

9) The authors performed cytotoxicity tests on three cancer lines. No information is reported on non-tumor line toxicity. The authors should have also selected a non-tumor cell line.

Author Response

The study “Microbial biosynthesis of chrysazin derivatives in recombinant Escherichia coli and their biological activities” is interesting but I have several doubts about some experiments.

Thank you very much for your positive comments.

1)Please, uniform anticancer in the text. At some point is present anticancer in other anti-cancer.

Answer: Thank you for your suggestion. All the “anti-cancer” has been changed to “anticancer”.

2)Please, rephrase “In this study, a recombinant Escherichia coli strain was developed to modify chrysazin” in “This study developed a recombinant Escherichia coli strain to modify chrysazin”.

Answer: Thank you for your suggestion. An appropriate correction has been duly made as “This study developed a recombinant Escherichia coli strain to modify chrysazin to chrysazin-8-O-α-L-rhamnoside (CR) and chrysazin-8-O-α-L-2′-O-methylrhamnoside (CRM) using rhamnosyl transferase and sugar-O-methyltransferase.”

3) please, uniform anti-microbial in the text. In some points is present antimicrobial in other anti-microbial.

 Answer: Thank you for your suggestion. All “anti-microbial” are changed to “antimicrobial”.

4)Please, change “Disc” to “disk”.

Answer: Thank you for your suggestion. An appropriate correction has been duly made.

5) Generally, the disk diffusion assay is performed following EUCAST or CLSI guidelines. Is not clear which kind of protocol the authors used. Please, add a reference.

 Answer: Thank you for your suggestion. We followed the Clinical and Laboratory Standards Institute (CLSI) guidelines and Kirby–Bauer method for the disk diffusion assay and appropriate references were added.

[32. Clinical and Laboratory Standards Institute, Performance standards for antimicrobial disk susceptibility tests: Approved standard - Eleventh edition, 2012. https://doi.org/M02-A11.

  1. Hudzicki, J. Kirby-Bauer Disk Diffusion Susceptibility Test Protocol Author. Am. Soc. Microbiol. 2012, 1–13. https://www.asm.org/Protocols/Kirby-Bauer-Disk-Diffusion-Susceptibility-Test-Pro.]

6) In the disk diffusion assay the authors used inocula containing 107 colony forming units (CFU)/mL. The guidelines recommend to use 0.5 McFarland which corresponds to 1.5 x10CFU/mL and measuring after 18-20h. Why in the method of micro dilutions and agar the result was evaluated at different times, 12 and 20h?

 Answer: Thank you for your suggestion. There was a typing mistake and we corrected it as ‘Inocula containing 1 × 108 colony forming units (CFU)/mL were spread onto MHA plates. Samples were then incubated at 37°C for 18-20 h.

7) Figure s5 is not visible. I can not appreciate the measures of the diameter area.

 Answer: Thank you for your suggestion. Figures were kept enlarged and presented in the Supplemental information section.

8)erythromycin was used as a positive control in all antibacterial tests. All MRSA strains were sensitive to this antibiotic? Are antibiograms available?

 Answer: Thank you for your suggestion. Not all the MRSA strains were sensitive to this antibiotic however, the antibiograms aren’t available.

9) The authors performed cytotoxicity tests on three cancer lines. No information is reported on non-tumor line toxicity. The authors should have also selected a non-tumor cell line.

 Answer: We tested on non-tumor cell lines and we found these compounds had a lower inhibitory effect on normal cells (Table S1 and Figure S6).

Table S1. Anticancer (IC50 (mM)) potential of chrysazin analogues against different cancer cell lines.

Cell Lines

Chrysazin (Chy)

CR

CRM

Cancer cell lines

AGS

17.08

28.58

7.513

Huh7

30.53

21.28

4.467

HL60

22.24

14.68

4.540

Normal cell lines

HaCaT

>200

>200

>200

Round 2

Reviewer 2 Report

Although the response on the sensitivity of MRSA strains to erythromycin was not complete, the authors answered the other questions well, so the manuscript could be accepted.